# The Influence of the Tool Tilt Angle on the Heat Generation and the Material Behavior in Friction Stir Welding (FSW)

Bahman Meyghani [1,2,]* and Mokhtar Awang [3,]*

1    BKL B.V., Collse Heide 1, 5674 VM Nuenen, The Netherlands
2    Department of Mechanical Engineering, Faculty of Engineering Technology and Built Environment, UCSI University, Taman Connaught, Kuala Lumpur 56000, Malaysia
3    Mechanical Engineering Department, Universiti Teknologi Petronas, Seri Iskandar 32610, Malaysia
*    Correspondence: bahman.meyghani@bkl.nl (B.M.); mokhtar_awang@utp.edu.my (M.A.)

**Abstract:** To improve the accuracy of numerical simulation of friction stir welding (FSW) process, the tool tilt angle must be considered as a significant parameter. In this study, specific considerations for mechanical boundary conditions in Eulerian domain is employed to investigate the tool tilt angle influence on the thermomechanical behavior in FSW. Aluminum 6061-T6 with a thickness of 6 mm under a rotational speed of 800 RPM, a transverse speed of 120 mm/min, and a plunging depth of 0.1 mm were employed for the simulations. Results showed an almost symmetric temperature profile predicted by the model without considering the tool tilt angle, while after incorporating the tool tilt angle, the peak temperature point is moved to the tool backside (around 400 °C), resulting in better material bonding, enhancing the weld joint quality. Without accounting for the tool tilt angle, the highest temperature of 389 °C is observed, while with the tilt angle the maximum temperature of 413 °C is achieved. The temperature variations at different points of the leading (around 360 °C) and the trailing sides (around 400 °C) of the welding tool were measured. It was observed that, after considering the tilt angle, as the tool moves, a smooth and quick increase for the temperature at the tool trailing side is achieved. This smooth and quick increasing of the temperature at the trailing side results in reducing the possibility of the formation of defects, cracks, and voids. Finally, comparisons showed that the model computational time is acceptable, and using Eulerian formulation leads to achieving a remarkable accuracy.

**Keywords:** tilt angle; Friction Stir Welding (FSW); shear layer; heat generation; material flow; finite element model; eulerian; thermomechanical; temperature

## 1. Introduction

As a solid-state joining method, friction stir welding (FSW) is used for joining different materials such as aluminium, magnesium, and steel together [1,2]. Investigating the thermal and mechanical behaviour of FSW is a very complicated task, because so many parameters such as tool geometry, process parameters, plunging depth, tool tilt angle, tool offset, etc. affect the intermediate variables of the FSW process [3].

All of the above-mentioned parameters highly affect the welding frictional behavior and thereby the welding quality. However, between the abovementioned parameters, the influence of the tool tilt angle (Figure 1) on the welding quality is not completely studied by the literature [4]. To illustrate, the presence of the tilt angle causes an additional forging force at the trailing side of the FSW [5]. This force results in better bonding of the material at the backside of the tool, thereby achieving a high quality and defect-free weld. Therefore, applying the tilt angle in practical FSW is mandatory [6]. On the other hand, the changes of the tool position after applying the tilt angle have an important influence on the geometry of the contact area and the material flow velocity [7,8]. Therefore, inappropriate tilt angle causes the formation of defects [9–11]. Hence, investigating the effect of the tilt angle on

the geometry of the contact area and the material flow velocity should be considered as significant issues [12].

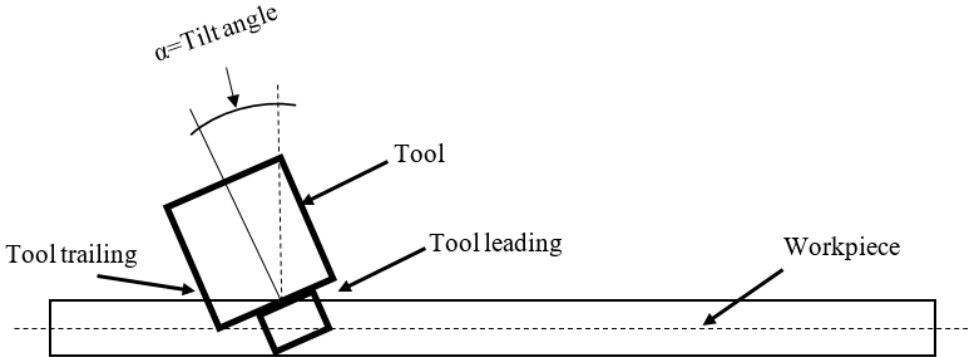

**Figure 1.** An exaggeratedly cross-sectional view of the workpiece and the tool at the longitudinal direction while the tilt angle is applied.

During the past decades, researchers [13–15] used experimental methods to investigate the influence of the tilt angle on the mechanical properties and welding characteristics. The results showed that increasing the tilt angle increases the torque and welding forces. A study [16] examined the effect of the tilt angle on the size and the location of the defects. The results showed that the optimum values of the tilt angle in the case of AA2219 is 1–2 degrees.

In recent years, finite element methods (FEMs) are assisting the researchers in investigating the tilt angle more in detail, because the experimental investigations are costly and time consuming [17]. Zhang et al. [18] studied the influence of the tilt angle on the heat and mass transfer of FSW in a computational solid mechanic (CSM) model. For finding the impact of the tilt angle on the geometry of the contact area, an expectation based on the experimental tests are conducted. Coulomb friction model is used for the interaction section. The results showed that tilt angle increases the welding temperature and the frictional force. It is claimed that these parameters affect the material flow velocity at the backside of the tool. Dialami et al. [19] studied the effect of the tilt angle on the temperature and material flow during FSW of AA2024-T4 using a computational fluid dynamic (CFD) model. The authors used constant values of the shear stress for modifying the Norton friction law. The results showed that, the material flow and the heat generation is enhanced after applying the tilt angle. It was claimed that the tilt angle has an important influence on the material flow velocity. Long et al. [20] applied different values of the tilt angle including 0, 1, and 2 degrees for welding of AA6061-T6 during different process parameters. A simple form for the geometry of the contact area was considered for the models. The results indicated that the prediction of the temperature in the model with the tilt angle of 2 degrees is higher than others. Furthermore, the material flow velocity was enhanced after applying the tilt angle. A paper [21] investigated the influence of the tilt angle of 1.5° and 3° on the material flow behavior. AA6082-T6 at different ranges of the rotational and transverse speeds, including 500 and 1000 RPM and 70, 100, and 150 mm/min was set for both the experiments and the numerical simulation. It was found that the flow of the metal and the temperature were scrutinized after applying different values of the tilt angle. Different values of the tilt angle including 2°–4° have been applied to the tool for FSW of Al2024-T3 by Meyghani et al. [22]. Analogous thermal profiles recorded by experiments approved as the base for the simulation model derivation, and it was claimed that ANSYS® model can provide accurate results. Therefore, the problem of the mesh distortion was solved due to the employment of a CFD based model. Zhai et al. [5] used a CFD-based model to investigate the effect of the tilt angle on the heat transfer and material flow. The results show that the forging force increases after applying the tilt angle. The temperature region becomes larger at the trailing advancing side. Furthermore, the material flow around the tool also becomes more intensified. It was also claimed that the tilt angle has an influence on the geometry of the contact area.

The review of the literature showed that the tilt angle has a significant influence on the intermediate variable of FSW process such as stress, temperature, the material plastic deformation, material flow velocity, etc. During the modeling of the process, the tilt angle makes the interaction behavior of the contact area, the material flow velocity, the distortion of the mesh, and the computational costs challenging. To overcome the challenges of the interaction behavior, a simplified Norton friction model based on the experimental observations was used for CFD models. However, because of human error, improper calibrations, measurement estimation, measurement device limitations, etc., there is always inaccuracy in experimental measurements. In addition, mesh distortion and computational costs were solved by using a Eulerian domain. In the Eulerian domain, the material flows through the faces of the element and the mesh is fixed in the space. Thus, the use of this flow boundary condition in the Eulerian method results in reducing the mesh distortion and also the simulation time significantly. Some CSM models showed a better performance in handling the mesh distortion and computational costs problems. However, due to the limitations in implementing the friction model in a Eulerian domain, they used a simplified frictional and the material flow velocity behaviors. This issue leads in achieving inconsistent results for the simulated model.

To solve the abovementioned problems, for the first time in this paper, a modified version of the Norton friction model in a Eulerian-based CSM model is implemented to simulate the FSW process. In the modified Norton model, a mathematically and theoretically based formulation is presented to investigate the influence of the tilt angle on the geometry of the contact area. Moreover, the presented friction model is able to consider the effect of the tool tilt angle on the material flow velocity. Finally, to investigate the influence of the tilt angle on the intermediate variable of the FSW, a comparison between the tilt angle of 0° and 1.5° in simulation and experimental study is performed.

## 2. Heat Generation Model Descriptions

Calculation of the temperature field during FSW can be performed by the following Equation (1) [18],

$$\frac{\partial}{\partial x}\left(k\frac{\partial T}{\partial x}\right) + \frac{\partial}{\partial y}\left(k\frac{\partial T}{\partial y}\right) + \frac{\partial}{\partial z}\left(k\frac{\partial T}{\partial z}\right) + \dot{q}_{pl} = \rho c_p \frac{\partial T}{\partial t} \tag{1}$$

where $T$ is the temperature, $t$ is the time, $\dot{q}_{pl}$ is the generated heat from the plastic deformation, $\rho$ is the material density, and $c_p$ is the specific heat.

As a source term, the heat from plastic deformation is directly available in Equation (1). The heat from friction can be applied as a boundary condition.

### 2.1. Heat from Friction

As mentioned, the heat from the friction can be applied as boundary condition for the model. As can be seen in Figure 2, the heat from friction is applied at the top surface of the workpiece [23].

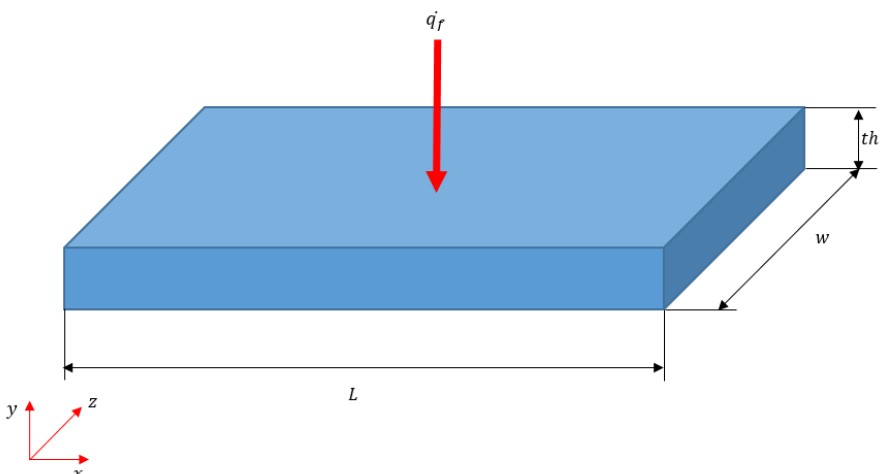

**Figure 2.** The schematic view of the workpiece and the generated heat by friction.

The heat from friction can be investigated by Equation (2) [18],

$$\dot{q}_f = (\Delta v_g)\tau \tag{2}$$

where $\dot{q}_f$ is the generated heat from the friction (w/m$^2$), $(\Delta v_g)$ is the relative velocity (m/s), and $\tau$ is the shear stress (N/m$^2$).

Therefore, the shear stress and the relative velocity should be investigated to calculate the heat from the friction.

### 2.1.1. Norton Friction Model

In this study, Norton friction law (Equation (3)) is employed to find the frictional behavior in the contact area [19].

$$\tau = -P \times g(\Delta v_g) \times \frac{\Delta v_g}{|\Delta v_g|} \tag{3}$$

where $\tau$ is the shear stress (N/m$^2$), $P$ is the pressure (N/m$^2$), and $g(\Delta v_g)$ is the defined function for the relative velocity between the tool velocity and the material velocity [24,25]. It should be noted that $g(\Delta v_g)$ is a dimensionless function as below [24,25].

The tilt angle mainly affects two significant parameters that are available in Equation (3): firstly, the pressure, because of its influence on the geometry of the contact area, and secondly, the material flow velocity.

### 2.1.2. The Influence of the Tilt Angle on the Contact Pressure

Equation (4) shows the general formulation for the pressure at the welding interface,

$$P = \frac{F}{A} \tag{4}$$

where $F$ is the welding force and $A$ is the contact area.

Figure 3 indicates that the tilt angle affects the geometry of the contact area. As can be seen, the calculation of the in-contact area after applying the tilt angle is a difficult task because its geometry is non-regular. In this paper, for the first time a mathematically based formulation is introduced for calculating the contact area after applying the tilt angle.

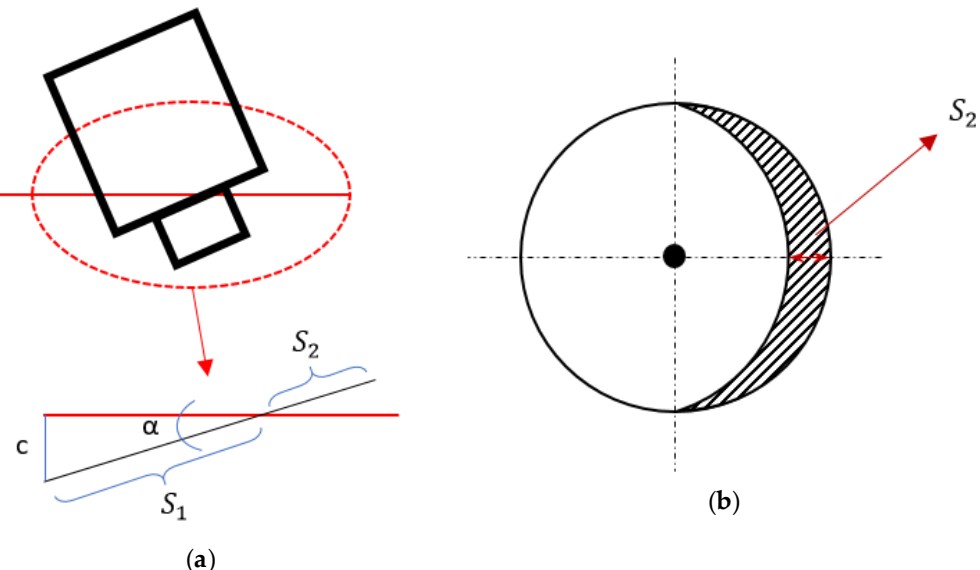

**Figure 3.** Schematic view of the tilt angle in (**a**) X-Y and (**b**) X-Z surfaces, c is the penetration depth of the trailing edge of the shoulder, $\alpha$ is the tool tilt angle, $S_1$ is the part of the shoulder, which is in contact with the workpiece, and $S_2$ is the part of the shoulder which is outside of the contact area.

As illustrated in Figure 4, in order to calculate the contact area, an additional circle (the diameter of both circles is the same as the shoulder) is added in the left side of the contact area (red circle). Therefore, an additional $S_2$ will be added to the left side of the contact area.

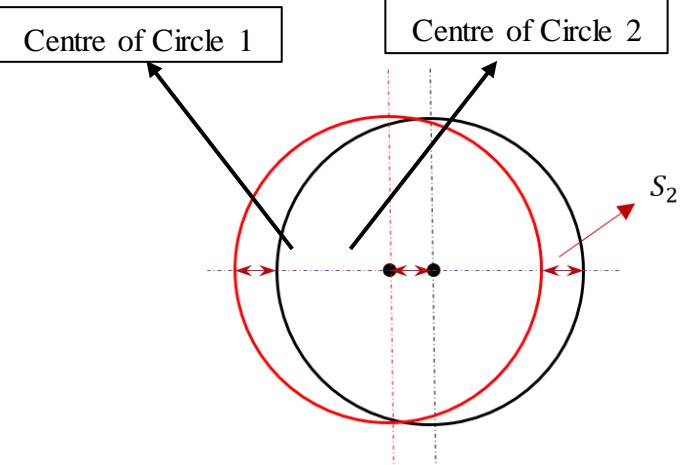

**Figure 4.** Schematic view from the top view for the in contact and not contact area including the additional circle.

As can be seen in Figure 5, two different parts for the contact area, for which one of them belongs to circle one and another one belongs to circle two, will be achieved. Each one these areas can be considered as an arc segment area.

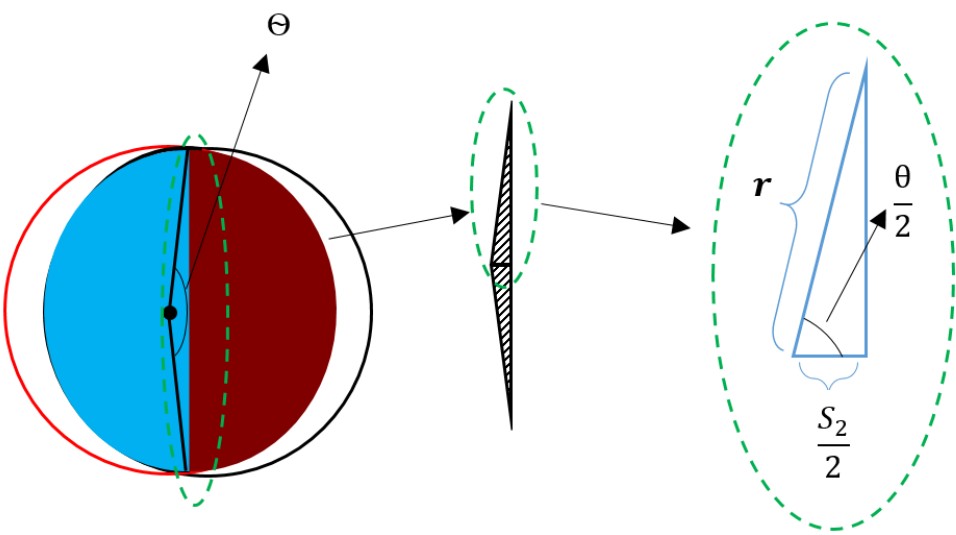

**Figure 5.** Schematic view of the circle and its relationship with the arc segment area.

In the Cartesian coordinates $x = (S_2/2)$, while in the polar coordinates $x = r\cos\theta$, therefore,

$$r\cos\theta = \frac{S_2}{2} \tag{5}$$

Thus,

$$r = \frac{S_2}{2\cos\theta} \tag{6}$$

The equation of a circle in polar coordinate is equal to $R$, and in Cartesian coordinates, it is equal to $x^2 + y^2 = R^2$, therefore $r$ value in Equations (5) and (6) is equal to $R$ in polar coordinates, therefore,

$$r = R = \frac{S_2}{2\cos\theta} \tag{7}$$

and

$$\theta = \cos^{-1}\left(\frac{S_2}{2R}\right) \tag{8}$$

From mathematics, the area under a curve can be calculated by using definite integration formulation. Hence, the area below the shoulder is equal to,

$$A = \frac{1}{2}\int_{-\cos^{-1}\left(\frac{S_2}{2R}\right)}^{\cos^{-1}\left(\frac{S_2}{2R}\right)}\left(R^2 - \frac{S_2^2}{(2\cos\theta)^2}\right)d\theta \tag{9}$$

According to the even functions rules we have,

$$f(\theta) = f(-\theta) \tag{10}$$

Thus,

$$A = \int_0^{\cos^{-1}\left(\frac{S_2}{2R}\right)}\left(R^2 - \frac{S_2}{(2\cos\theta)^2}\right)d\theta \tag{11}$$

$$A = \frac{R^2 - S_2^2\tan\theta}{4}\bigg|_0^{\cos^{-1}\left(\frac{S_2}{2R}\right)} = R^2\cos^{-1}\left(\frac{S_2}{2R}\right) - \frac{S_2^2}{4}\tan(\cos^{-1}\left(\frac{S_2}{2R}\right)) \tag{12}$$

$$A = 2\left[R^2\cos^{-1}\left(\frac{S_2}{2R}\right) - \frac{S_2^2}{4}\tan(\cos^{-1}\left(\frac{S_2}{2R}\right))\right] \tag{13}$$

It should be noted that the additional number 2 is added because of the presence of the left and the right sides of the in-contact area (blue and crimson circles in Figure 6). The abovementioned formulation gives the calculated results of contact area.

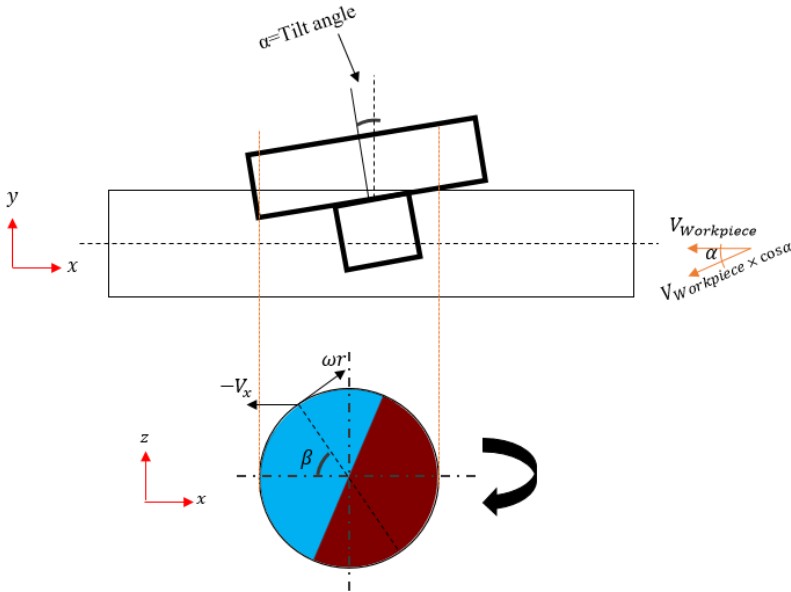

**Figure 6.** Rotation of the contact area.

Therefore, the pressure at the contact area after applying the tilt angle can be calculated from the below equation,

$$-P = \frac{F}{2\left[R^2 \cos^{-1}\left(\frac{S_2}{2R}\right) - \frac{S_2^2}{4} \tan(\cos^{-1}\left(\frac{S_2}{2R}\right))\right]} \tag{14}$$

2.1.3. The Influence of the Tilt Angle on the Relative Velocity

The material flow velocity can be investigated from Equation (15),

$$V_R \times r_{Shoulder} = Material\ flow\ velocity\ \left(\frac{m}{s}\right) \tag{15}$$

where $V_R$ is the tool rotational speed (radian/s) and $r_{Shoulder}$ (m) is the shoulder radius.

It should be mentioned that the literature [5,18,19] claimed that, after applying the tilt angle, the contact area also rotates across the x-direction (Figure 6). This rotation does not affect the geometry of the contact area; however, according to Equation (16) to Equation (18), it significantly affects the material flow velocity.

Therefore, after applying the tilt angle, the components of the material flow velocity can be explained as below,

$$V_x = V_R \cdot r_{Shoulder} \cdot \sin\beta \cdot \cos\alpha \tag{16}$$

$$V_y = V_R \cdot r_{Shoulder} \cdot \cos\beta \tag{17}$$

$$V_z = -V_R \cdot r_{Shoulder} \cdot \sin\beta \cdot \sin\alpha \tag{18}$$

where $V_x$, $V_y$, and $V_z$ are the components of the material flow velocity in $x, y$, and $z$ directions, and $\beta$ is the rotational angle relative to the x-direction.

### 2.2. Heat from Plastic Deformation

The generated heat from the plastic deformation during FSW is explained as below [18],

$$\dot{q}_{pl} = \eta \sigma : \dot{\varepsilon}_{pl} \tag{19}$$

where $q_{pl}$ is the generated heat from the plastic deformation (heat flux per unit volume (w/m$^3$)), $\sigma$ is stress (N/m$^2$), $\eta$ is the inelastic heat fraction (input parameter) which is dimensionless, and $\dot{\varepsilon}_{pl}$ is the plastic strain (1/s).

To solve Equation (19), the stress and the plastic strains should be calculated.

#### 2.2.1. Stress Calculation

In order to calculate the total amount of the stress ($\sigma$), the true strain ($\varepsilon$) (input parameter) should be multiplied by the stiffness matrix ($D$) as follows,

$$\boldsymbol{\sigma} = D : \varepsilon \tag{20}$$

The tensor form of Equation (20) is written as,

$$\boldsymbol{\sigma}_{ij} = D_{ij \times l} \times \varepsilon_l \tag{21}$$

and the matrix form of Equation (20) can be written as,

$$
\begin{bmatrix} \sigma_{11} \\ \sigma_{22} \\ \sigma_{33} \\ \sigma_{12} \\ \sigma_{13} \\ \sigma_{23} \end{bmatrix} =
\begin{bmatrix}
D_{1111} & D_{1122} & D_{1133} & D_{1112} & D_{1113} & D_{1123} \\
D_{2211} & D_{2222} & D_{2233} & D_{2212} & D_{2213} & D_{2223} \\
D_{3322} & D_{3322} & D_{3333} & D_{3312} & D_{3313} & D_{3323} \\
D_{1211} & D_{1222} & D_{1233} & D_{1212} & D_{1213} & D_{1223} \\
D_{1311} & D_{1322} & D_{1333} & D_{1312} & D_{1313} & D_{1323} \\
D_{2311} & D_{2322} & D_{2333} & D_{2312} & D_{2313} & D_{2323}
\end{bmatrix}
\times
\begin{bmatrix} \varepsilon_{11} \\ \varepsilon_{22} \\ \varepsilon_{33} \\ \varepsilon_{12} \\ \varepsilon_{13} \\ \varepsilon_{23} \end{bmatrix}
$$

After calculating the total amount of the stress, the hydrostatic pressure ($P \ (\text{N/m}^2)$) can be calculated as follows,

$$P = -\sigma_{Hyd} = -\frac{\sigma_{11} + \sigma_{21} + \sigma_{33}}{3} \tag{22}$$

Based on the von Mises yield criterion, the deviatoric stress is one the most important reasons for failure. This is caused due to the relationship between the hydrostatic stress and the volume change, whereby the deviatoric stress also has a relationship with the shape change.

The amount of the deviatoric stress can be found as below,

$$S = \sigma + PI \tag{23}$$

where $S$ is the stress deviatoric tensor $(\text{N/m}^2)$, $\sigma$ is the stress tensor (calculated from Equation (20)), $P$ is the hydrostatic pressure (can be investigated for Equation (22)), and $I$ is the identity tensor. Hence, it can be proved that the deviatoric stress and the stress have a relationship with the shape change.

#### 2.2.2. Plastic Strain Calculation

To figure out the plastic strain, the deviatoric stress, which is explained in the previous section, should be implemented in the von Mises yield criterion as follows,

$$\sigma_v = \sqrt{\frac{3}{2} S : S} \tag{24}$$

where $\sigma_v$ is the equivalent tensile stress.

If $\sigma_v$ becomes smaller that the material yield criterion, the total strain can be found as follows,

$$\Delta\varepsilon = \Delta\varepsilon^{el} \tag{25}$$

where $\Delta\varepsilon$ is the total value for the stress and $\Delta\varepsilon^{el}$ is the amount of the elastic strain.

If the yield criterion becomes larger than the material yield criterion, the material elastic phase will be changed to the plastic phase. As a result, the total plastic strain can be calculated from the following equation,

$$\Delta\varepsilon = \Delta\varepsilon^{el} + \Delta\varepsilon^{pl} \tag{26}$$

where $\Delta\varepsilon^{pl}$ is the plastic strain.

Finally, $\dot{q}_{pl}$ can be investigated.

Consequently, by finding the heat from the friction (Equation (2)) and the plastic deformation (Equation (19)), the total amount of the heat (Equation (1)) can be found.

### 3. Finite Element Model Descriptions

A 3D FEM model is established to investigate the temperature and material flow. A higher quality mesh is implemented to represent the results inside the heat-affected zone (HAZ) as high temperature gradient is supposed to be achieved close to the welding nugget zone (Figures 7 and 8). The diameter of the shoulder is 15 mm, and the pin diameter of 6 mm is considered for the model. A 6 mm is made for the pin length. The workpiece has a dimension of 200 mm (length) × 75 mm (width) and the thickness of 6 mm. The rotational velocity of 800 RPM and the transverse speed of 120 mm/min are selected for the welding. It should be noted that in the literature, these values are described as the optimize process parameters [4,17].

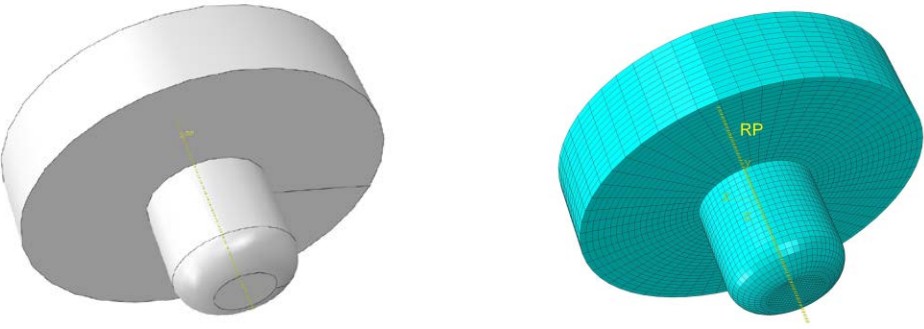

**Figure 7.** The welding tool including the part and the mesh.

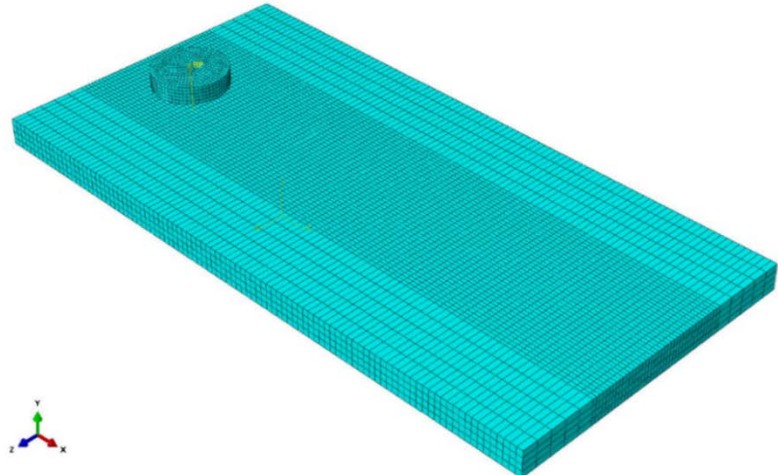

**Figure 8.** Schematic view of the assembled model.

### 3.1. Plasticity Modelling

Johnson and Cook [26] material law is applied to the model in order to simulate the elastic–plastic behavior.

$$\sigma_y = \left[ A + B(\varepsilon_P)^n \right] \left[ 1 + C \left[ \frac{\dot{\varepsilon}_P}{\dot{\varepsilon}_0} \right] \right] \left[ 1 - \left[ \frac{T_{FSW} - T_{room}}{T_{melt} - T_{room}} \right]^m \right] \tag{27}$$

where $\varepsilon_P$, $\dot{\varepsilon}_P$, and $\dot{\varepsilon}_0$ represent the effective plastic strain, the effective plastic strain rate, and normalizing strain rate, $T_{room}$ is the room temperature, $T_{melt}$ is the material melting temperature, $A$ represents the yield stress, $B$ represents the strain factor, $n$ represents the strain exponent, $C$ represents the strain rate factor, and $m$ represents the temperature exponent.

Moreover, all clamped portions of the plates constrained in all directions [27]. The workpiece entire bottom nodes are constrained in the perpendicular direction to emulate the support at the bottom of the plates. The loading comprises of three primary phases including plunging, dwelling, transverse, and plunging out. Tables 1–3 indicate the material properties.

**Table 1.** The properties and the parameters of the Johnson–Cook material law.

| Parameter | Explanation |
|:---:|:---:|
| $A$ | Yield stress |
| $B$ | Strain factor |
| $n$ | Strain exponent |
| $C$ | Strain rate factor |
| $m$ | Temperature exponent |
| $\varepsilon_P$ | The effective plastic strain |
| $\dot{\varepsilon}_P$ | The effective plastic strain rate |
| $\dot{\varepsilon}_0$ | Normalizing strain rate |
| $T_{room}$ | Room temperature |
| $T_{melt}$ | Material melting temperature |

**Table 2.** AA 6061-T6 properties (temperature-dependent).

| Temperature (°C) | Elasticity Modulus ($E$) | Poisson's Ratio [ν] | Thermal Expansion Coefficient | Specific Heat (J/Kg °C) |
|:---:|:---:|:---:|:---:|:---:|
| 37.8 | 69.7 | 0.3 | $2.345 \times 10^{-5}$ | 95 |
| 93.3 | 66.2 | 0.3 | $2.461 \times 10^{-5}$ | 978 |
| 148.9 | 62.7 | 0.3 | $2.567 \times 10^{-5}$ | 1004 |
| 204.4 | 59.2 | 0.3 | $2.669 \times 10^{-5}$ | 1028 |
| 260 | 53.49 | 0.3 | $2.756 \times 10^{-5}$ | 1052 |
| 315.6 | 47.78 | 0.3 | $2.853 \times 10^{-5}$ | 1078 |
| 371.1 | 39.75 | 0.3 | $2.957 \times 10^{-5}$ | 1104 |
| 426.7 | 31.72 | 0.3 | $3.071 \times 10^{-5}$ | 1133 |

**Table 3.** AA6061-T6 thermal conductivity properties (temperature dependent).

| Temperature (°C) | Thermal Conductivity (W/mK) |
|:---:|:---:|
| 148.9 | 162 |
| 204.4 | 177 |
| 260 | 184 |
| 315.16 | 192 |
| 371.1 | 201 |
| 426.7 | 207 |
| 148.9 | 217 |
| 204.4 | 223 |

### 3.2. Material Definition Descriptions

Tables 2 and 3 explain the thermal and mechanical properties of the AA-6061-T6. It should be noted that the temperature-dependent material properties are gained from the literature [28].

## 4. Results and Discussion

### 4.1. Stress Distributions

Figure 9 shows the distribution of the stress before and after applying the tilt angle. As can be seen in Figure 9a, the maximum stress is concentrated at the welding seam. However, the tilt angle caused an almost asymmetrical behavior for the stress distribution. As highlighted in Figure 9b, the tilt angle produced a rotation of the stress distribution trending to the advancing side. Moreover, the stress values for the model with the tilt angle at the step time of 0.4 and 0.8 s is around 257 and 356 MPa, respectively. After applying the tilt angle, these values increase to 405 and 426 MPa. The abovementioned issues are caused because the tilt angle increases the forces at the rear part of the advancing side, thus the maximum stress moves from the center of the welding to the advancing side. Moreover, this movement facilities the material softening and decrease of the material flow stress. This result agrees well with the presented results of the literature [29].

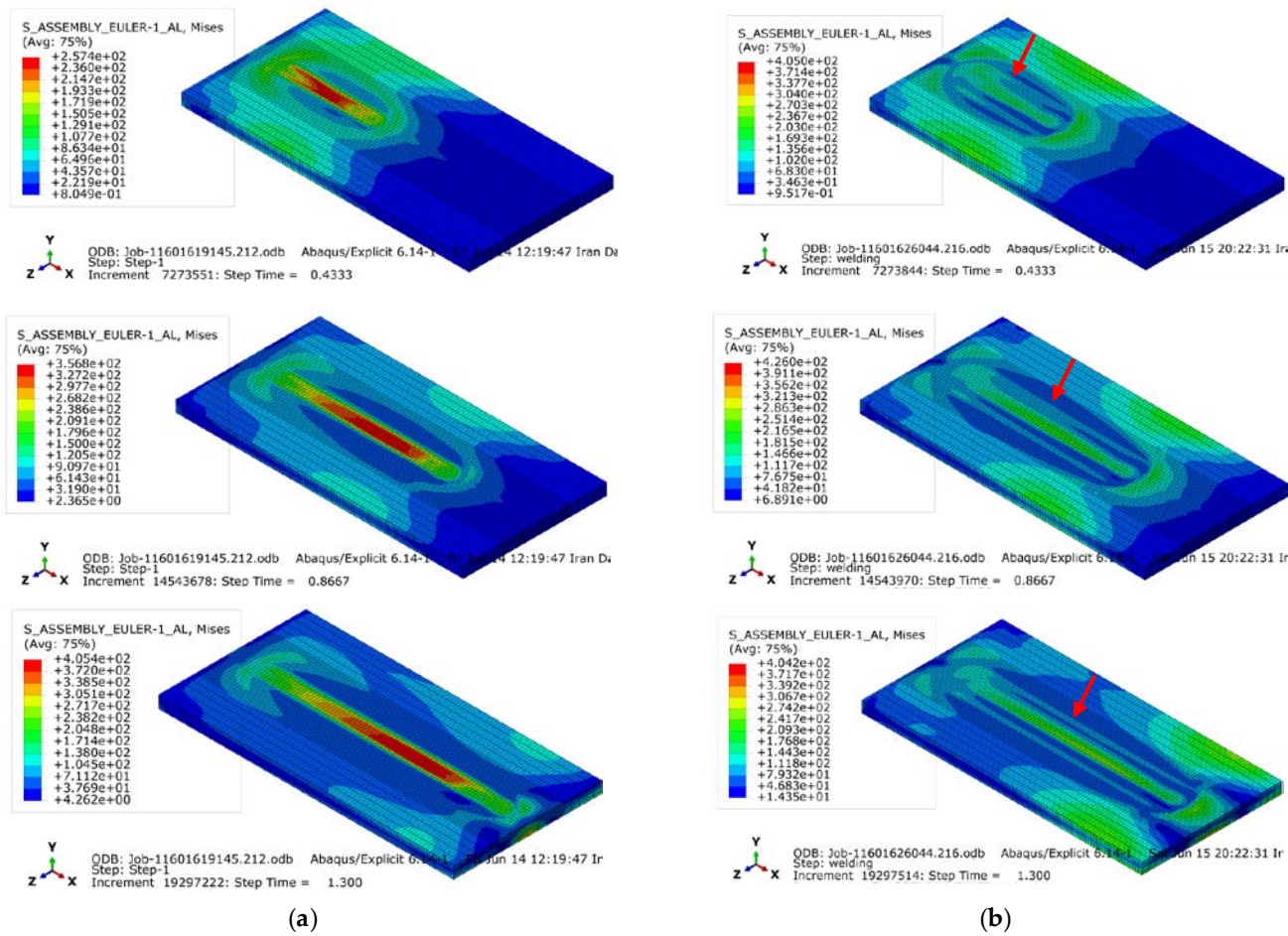

**Figure 9.** The distribution of the stress at different welding step times (**a**) without tilt angle (**b**) with tilt angle.

Figure 10 indicates the cross-sectional view of the stress distribution for the finite element models. As can be seen, at the center of the welding both models show an almost similar behavior. As the distance from the welding center line in the model in which the tilt angle of 1.5° increases, a more compressive stress behavior is observed.

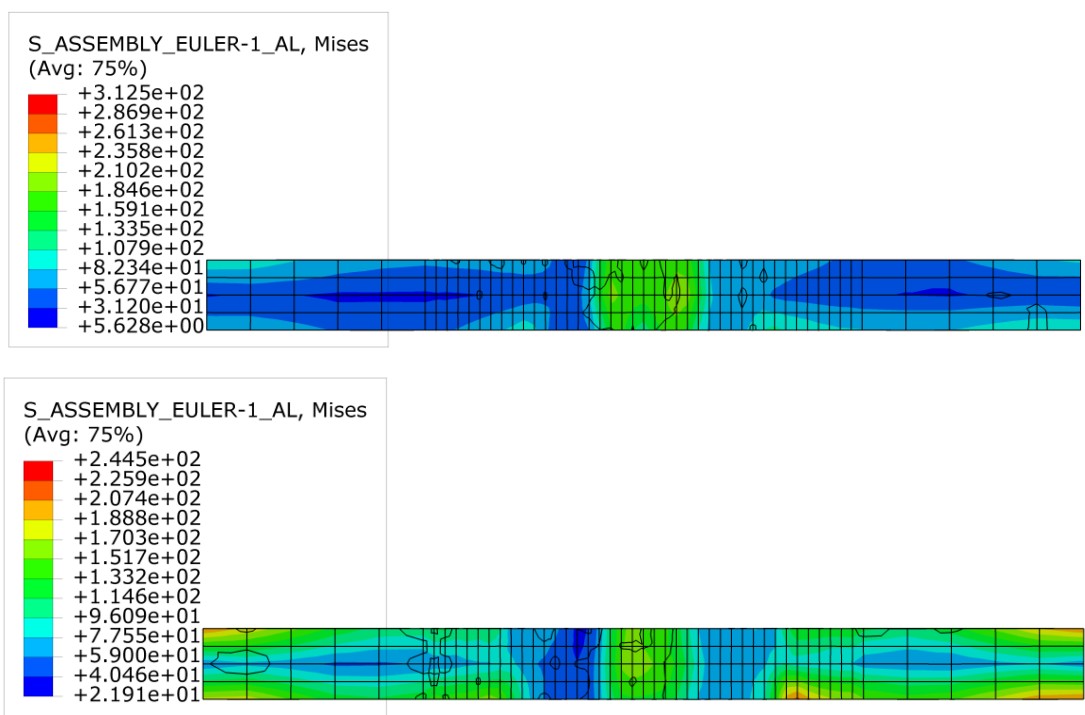

**Figure 10.** The distribution of the stress at the cross section.

Figure 11 shows the values of the stress distribution at the cross-sectional viewpoint. As can be seen, there is an almost "M"-shaped pattern for the stress in both modes. Figure 11 shows that, after applying the tilt angle, the stress values of 158 MPa at the advancing side (at the shoulder edges) and the maximum values of 150 Mpa at the retreating side is obtained. Before applying the tilt angle at the advancing side, the maximum stress of 130 MPa is observed. This is related to the higher shear force between the shoulder and workpiece especially at the string zone (SZ) [30]. Moreover, after applying the tilt angle, a smooth behavior for the vales of the stress is observed, while before applying the tilt angle, the stress values sharply increased from 40 MPa to 130 MPa. The same behavior for the values and the behavior of the stress is also reported in the literature [31,32].

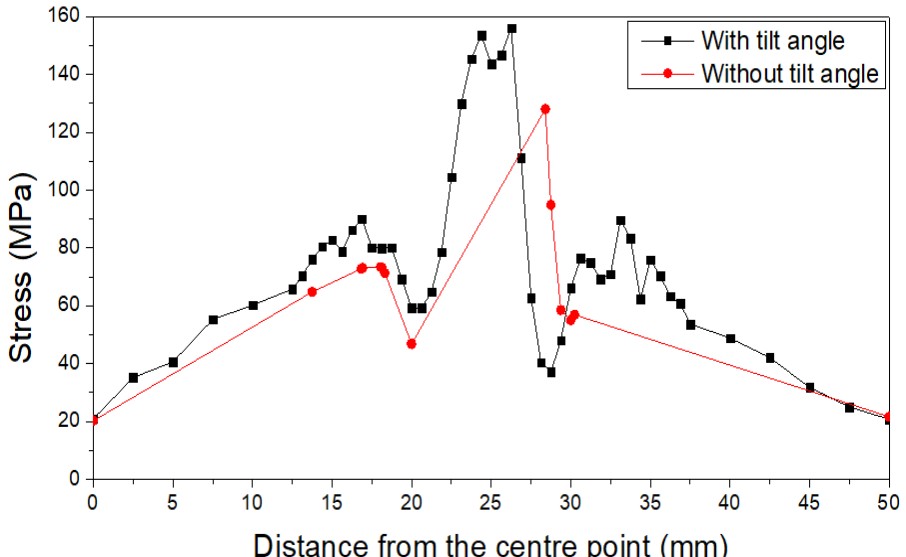

**Figure 11.** The cross-sectional view of the stress.

## 4.2. Heat Flux

Figure 12 shows the distribution of the heat flux in both models. It is clear that the tilt angle has a significant role in the shape and values of the heat flux. To explain more, after applying the tilt angle the heat flux becomes wider, because the increase in the welding forces especially at the back side of the tool.

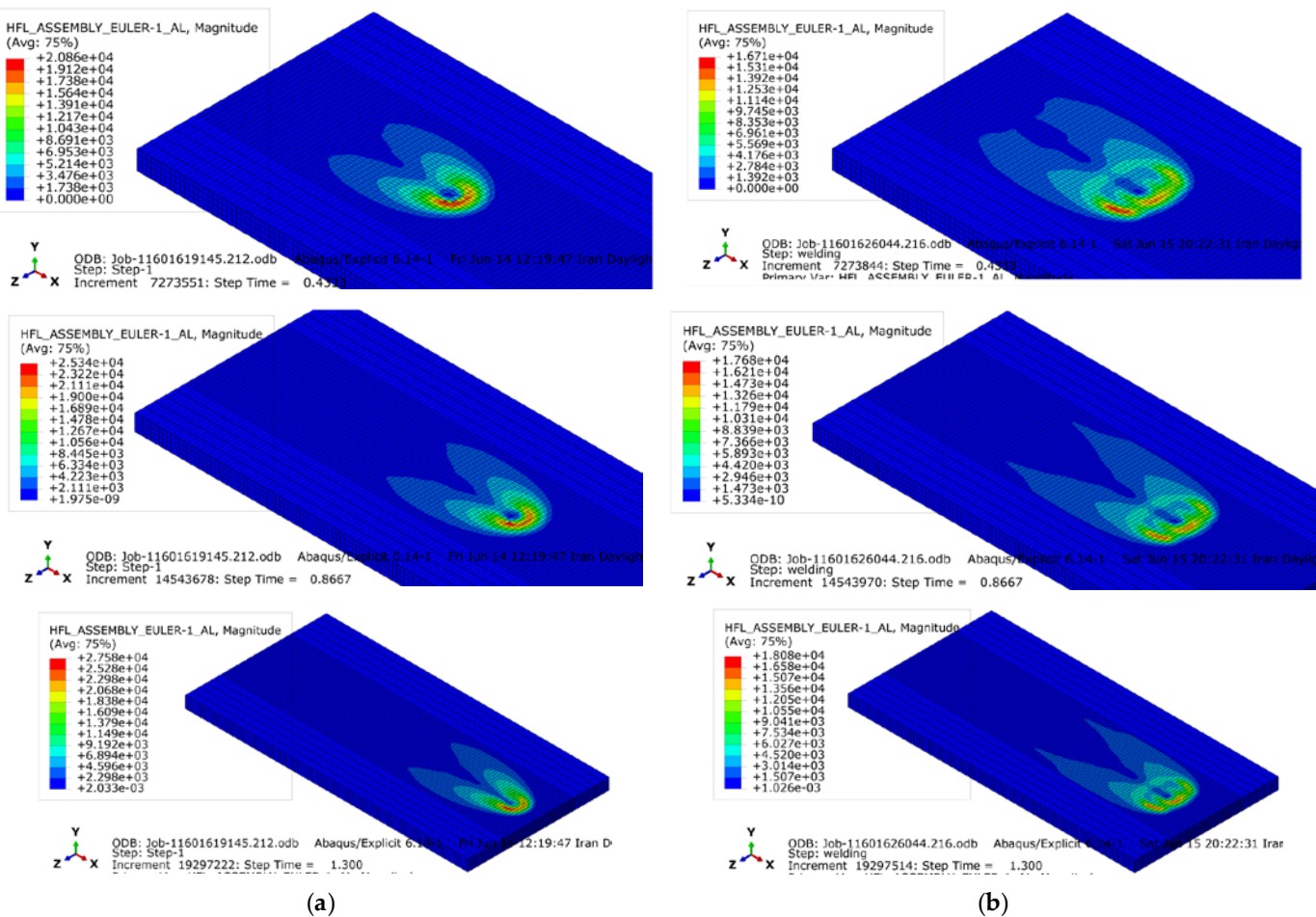

|     |     |
| --- | --- |
| (**a**) | (**b**) |

**Figure 12.** The distribution of the heat flux before (**a**) and after (**b**) applying the tilt angle.

Moreover, there is a difference at the center of the welding. The welding zone has a circle shape in the model with the tilt angle of 0°. This shape is converted to an oval one when the tilt angle of 1.5° is changed the tool position. This oval shape can confirm achieving a defect free weld. The literature [33] also reported that the presence of the tilt angle can decrease the possibility of the presence of a defect. Moreover, the maximum value of the heat flux is rotated and shaped an almost asymmetrical shape in the model with the tilt angle of 1.5°. In contrast, a symmetrical form for the heat flux is shaped in the model without the tilt angle (Figure 12a). Furthermore, at the ending of the welding step (step time 1.3 s), the complete circle located at the welding zone is converted to an incomplete circle. This issue may cause the presence of a defect at the welding seam.

## 4.3. Equivalent Plastic Strain (PEEQ)

The equivalent plastic strain for both models at different step times are plotted in Figure 13. The equivalent plastic strain shows a scalar variable, which indicates the material inelastic deformation. If the equivalent plastic strain becomes larger than zero, the material would be yielded. Figure 13a shows that without the tilt angle, the plastic deformation has been almost concentrated at the pin interface. However, Figure 13b shows wider values of the plastic strain, which shows a better string of the material especially at the shoulder area.

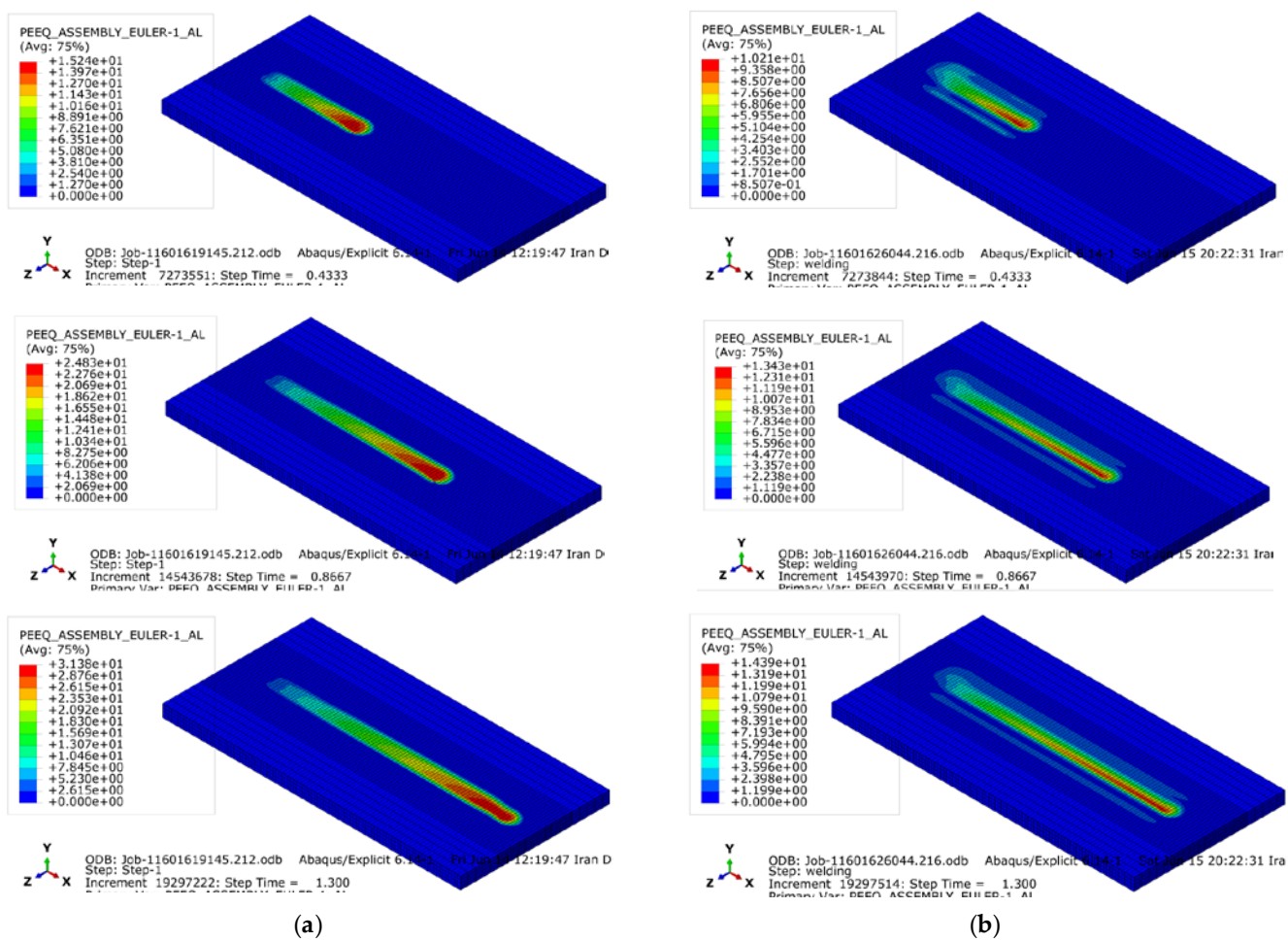

**Figure 13.** Three-dimensional view of the equivalent plastic strain (**a**) without the tilt angle, and (**b**) with the tilt angle.

As can be seen, like the stress and heat flux contour plots, there is an asymmetrical behavior for the equivalent plastic strain in the model with the tilt angle. Moreover, the rotational of the welding contact is also seen in the equivalent plastic strain results. The results of the plastic strain are in a good agreement with the reported values in the literature [34,35].

Figure 14 shows the longitudinal view of the equivalent plastic strain during the welding. As can be seen, a better bonding of the material at the pin bottom area is happening in the model with the tilt angle (Figure 14b). In contrast, the deformation of the material at the pin root in the model without the tilt angle has not been completely done. This issue indicates a failure in the disposition of the material, which may lead to the formation of a void.

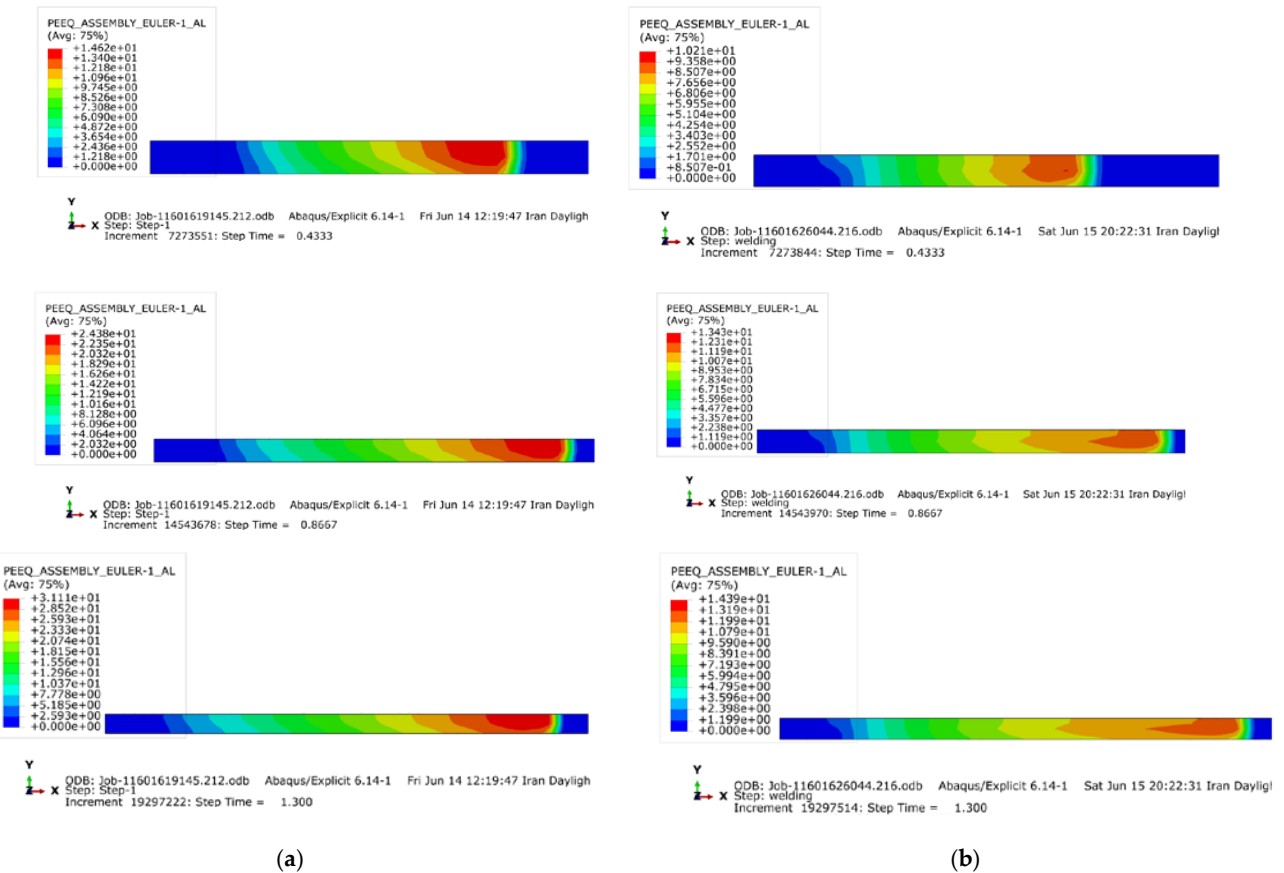

(**a**)               (**b**)

**Figure 14.** The longitudinal view of the equivalent plastic strain (**a**) with tilt Angele; (**b**) without the tilt angle.

The cross sectional of the equivalent plastic strain for both models can be seen in Figure 15. It is clear that there is a cylindrical burr shape for the plastic strain in both models. It was seen that the cylindrical burr shape at the cross section is bit larger than the pin diameter. The asymmetrical form is also seen at the cross-sectional view. The results for the cross-sectional view is confirmed with the literature [36].

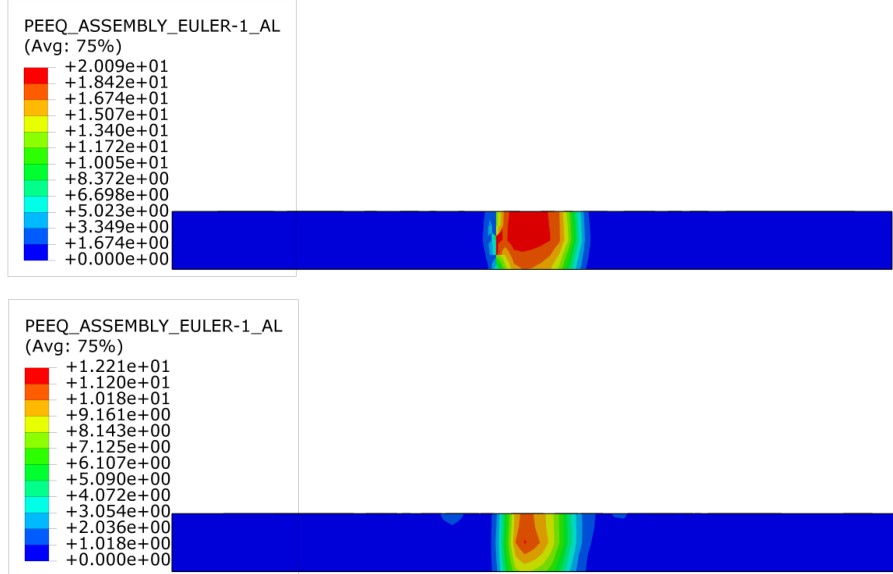

**Figure 15.** The equivalent plastic strain at the cross-sectional view of the welding.

### 4.4. Flow Velocity

The top view of the material flow velocity streamlines around the tool under two different conditions of the tilt angle are shown in Figure 16. It is found from the results of both models that the outside of the shoulder area, the transportation of the material mainly takes place at the retreating side. This asymmetrical behavior is happening because of an unbalanced behavior of the forces at different sides of the welding. Moreover, the expansion of the flow region especially at the retreating side gets larger after applying the tilt angle. In contrast, at the advancing side, a flow reversal or a stagnant zone can be seen. The abovementioned observations are in line with the reported results of the literature [5,37]. Figure 16b shows that the density of the streamlines is increased significantly.

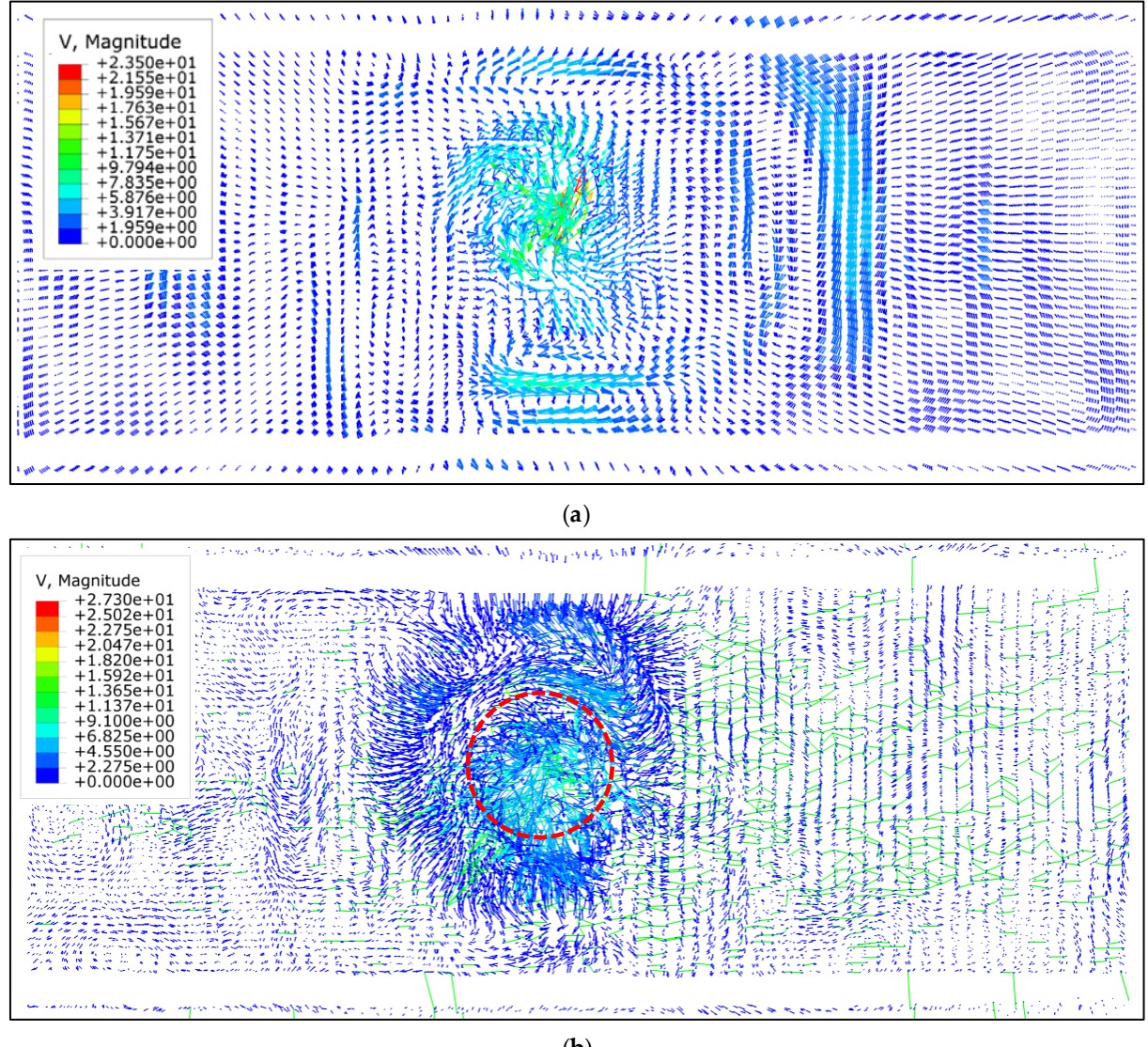

(**a**)

(**b**)

**Figure 16.** The material flow velocity (**a**) without the tilt angle, and (**b**) with the tilt angle.

The maximum value of the material flow velocity for the model with the tilt angle of 1.5 is found to be around 27 (m/s) which is almost equal to 30 percent of the tool velocity. The value is bit smaller (23 m/s) when the tilt angle is removed from the welding (Figure 16a). It can be also observed that, as the distance from the welding center line increases, the material velocity decreases.

### 4.5. Temperature Behavior

It needs to be mentioned that at the initial step of the welding, the temperature is fixed at room temperature (25 °C). Immediately, after touching the workpiece by the tool, due to the friction between the interfaces, the heat will be generated at the tool–material contact interface. After deforming the material during the plunging and dwelling stages, because of the plastic deformation and simultaneous dissipation of the generated heat occurring at some parts of the workpiece, the material deformation will be added as another source for the heat generation. During the welding, the amount of the generated heat is more than the amount of the heat that will be dissipated, and this issue causes a rise in the temperature at the welding center point. It should be noted that, during the coupled temperature displacement modelling, the quantity of the friction-generated heat is directly determined during the welding and measuring at each step of the simulation. In addition, the simulation continues until the establishment of a steady state propagating heat source and the extraction of the temperature profile from the center and behind the welding center point.

Figure 17 explains the distribution of the temperature before and after applying the tilt angle in both models. Figure 17a shows that, before applying the tilt angle, the temperature at the tool interface is found to be around 873 °C. From Figure 17c, it can be seen that as the generated heat is balanced with the heat dissipated, the temperature at the welding center point reaches the highest point (389 °C). Calculating the temperature at the interface of the tool and the distribution of the temperature at the adjacent places of the welding joint is made while the welding tool with 1.5° tilt angle is set (Figure 17b,d). Figure 17b,d indicates that the temperature at the welding tool is about 1118 °C, at the welding zone is around 413 °C, and 3 mm away from the welding is around 300 °C. As can be seen, the value of the temperature in this model is slightly more than the model without the tilt angle. The literature [19] reported that, due to the enhanced temperature and strain behaviors, after applying the tilt angle the mixing of the material will be improved, therefore the welding quality should be better, and the possibility of the defect will be decreased. To illustrate more, as the pin penetrates into the sheets, a movement for the volumetric heat source region occurs. Under these conditions, the material remains solid, thus no welding defects will be formed. Furthermore, higher generated heat in the back side of the weld assists in better material bonding and full embodiment of the material. Hence, it can be predicted that the presented model with the tilt angle is a free defect model.

Figure 17c,d indicates that in the model with the tilt angle, higher rate in the temperature change is found along the welding transverse line compared to the model without the tilt angle. This phenomenon can be explained by the fact that higher frictional force (produced by the tilt angle) occurred along the axis of the transverse movement. It can also be obtained that, after considering the tilt angle, the colored contour plot is larger than the model without the tilt angle. Furthermore, from the top section of the workpiece, the localized heating zone has a "mitosis cell division" shape (Figure 17d).

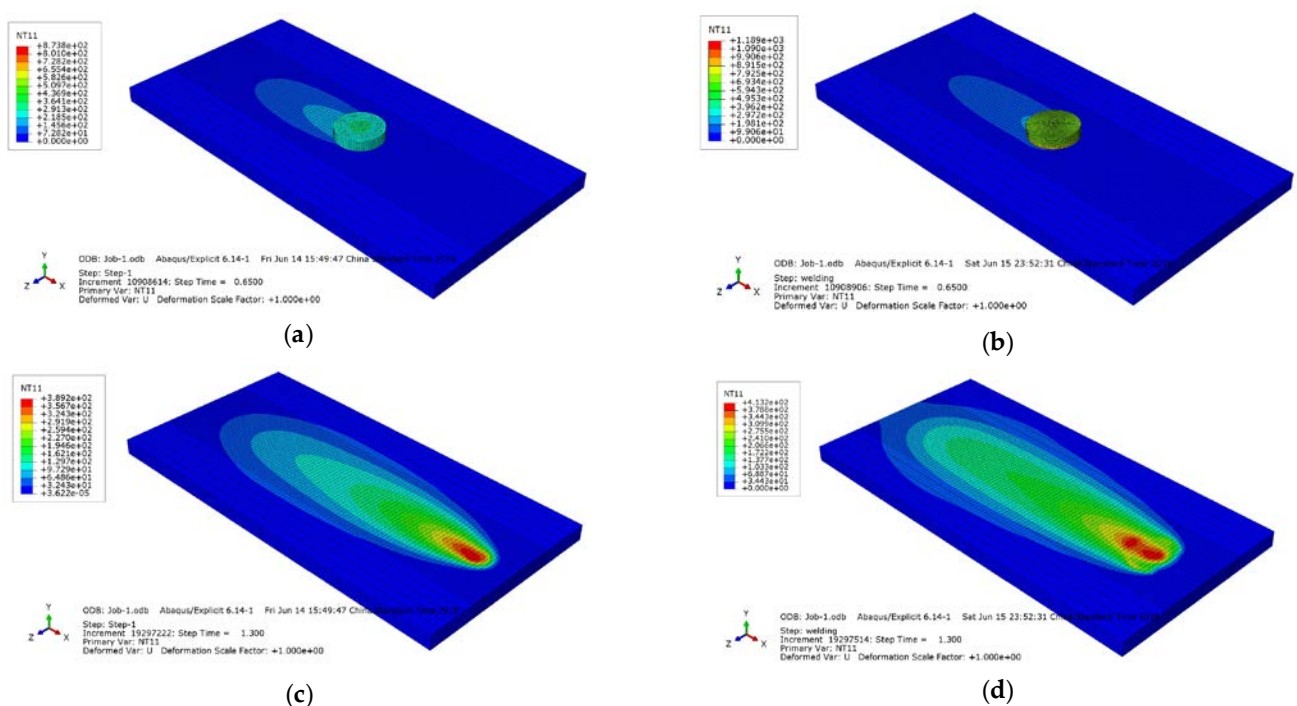

**Figure 17.** The temperature distribution at the welding surface in the model (**a**) with the tilt angle including the tool, (**b**) without the tilt angle including the tool, (**c**) with the tilt angle step time 1.3 s, and (**d**) without the tilt angle step time 1.3 s.

As seen in Figure 18a,c, before applying the tilt angle, the temperature at the retreating side is slightly lower than the advancing side, thus unlike the front and back side of the welding tool, an almost non-symmetrical temperature distribution is seen in the cross section. It should be noted that, although the peak temperature point is on the advancing side, its position is very close to the welding center line. In contrast, the cross section of the model with the tilt angle shows an almost symmetrical behavior at different sides of the tool (Figure 18b) and the front and the back sides of the tool (Figure 18b,d). The compassion between the results of this study and the literature [34] in which a symmetric shape for the shear layer and whereby the temperature behavior is considered are in line.

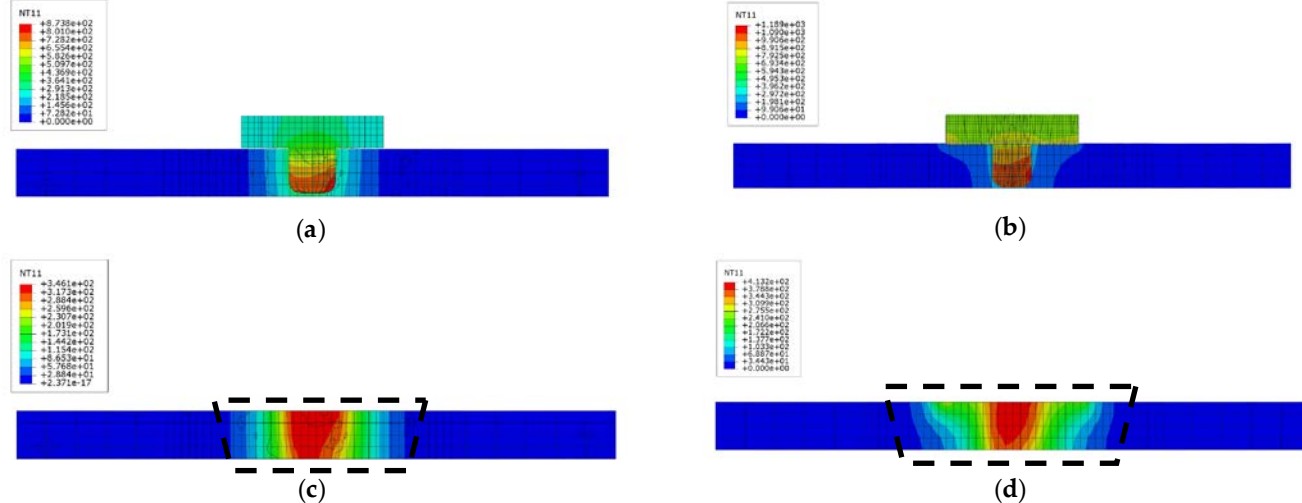

**Figure 18.** Temperature at the cross section with and without the tool (without tilt angle) (**a**) tool and tilt angle; (**b**) tool without the tilt angle; (**c**) cross section with the tilt angle; (**d**) cross section without the tilt angle.

At the cross section of both models, the region at the shoulder/workpiece interface experiences higher temperature ranges, because the majority of the produced temperature is detected in the shoulder region compared to the temperature at the pin region. This issue is caused because of the larger geometry of the shoulder, thereby larger contact interface area at the upper surfaces of the intersection between the workpiece and the tool shoulder. Thus, an almost "V"-shaped pattern for the temperature is formed at the cross section.

Due to the higher generated heat at the maximum tool/workpiece connection point (highlighted in Figure 19) in both models, the peak temperature is found at the tool back side region. The location of the peak temperature at the back side of the pin is also reported by the previous literature [34,38]. This issue shows the good agreement between the results of this study and those found in the previous literature. To explain more in detail, in the model with the tilt angle, the temperature at the back side of the tool is decreasing smoothly, while in the model without the tilt angle there is a sharp reduction in the values of the temperature (Figure 19b). The graph also confirms that the total generated heat in the model with the tilt angle is higher than the model without the tilt angle.

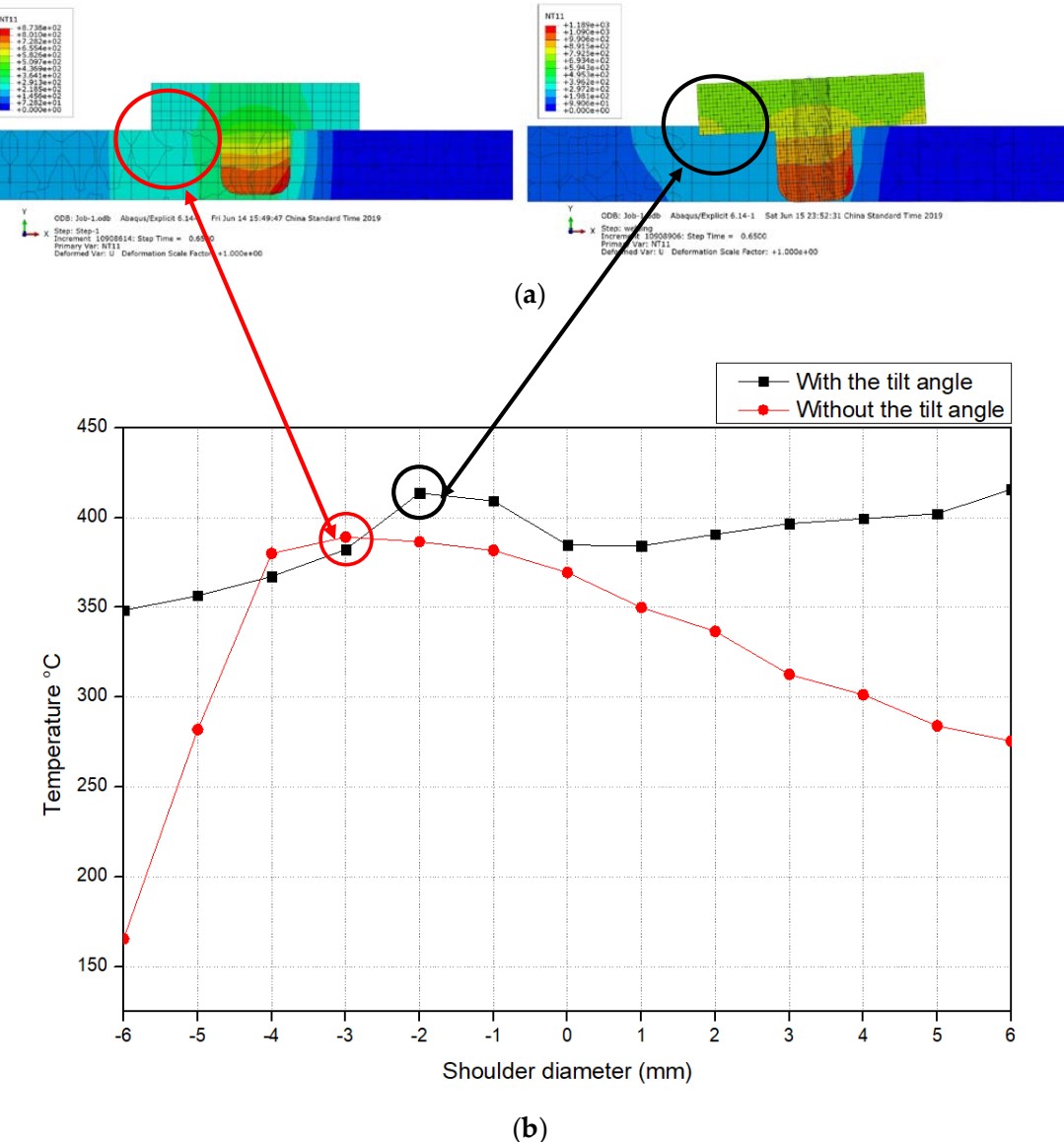

**Figure 19.** Temperature of the welding across the shoulder diameter (**a**) simulation results; (**b**) temperature results.

Another significant issue that needs to be considered is preheating performed by the welding tool, because it has an influence on the formation of cracks and voids inside the welding zone. In order to simplify the model in the literature [34,38] a uniform behavior for this preheating is considered for different models with and without the tilt angle. Figure 19 shows that preheating caused by the welding tool is higher in the model with the tilt angle. To illustrate the issue, due to the larger geometry of the contact area and larger values of the frictional force, the preheating in the model with the tilt angle is higher. Therefore, higher values of the preheating in the model with the tilt angle indicates that the accuracy of the results of this study is higher than the presented models in the literature [34,38]. Moreover, Figure 19b indicates that there is a smooth preheating behavior in the model with the tilt angle which leads to the uniform formation of the microstructure, while in the model without the tilt angle, the temperature is increased sharply which results in the formation of defects inside the welding zone. Thus, it can be predicted that the welding quality in the model with the tilt angle is higher.

The temperature curve across the cross section of the welding in both models (with and without tilt angle) along with the experimental results of the literature [39], and published literature is shown in Figure 20. As shown, the maximum value of the temperature is located inside the welding zone and the temperature decreases gradually through the heat affected zone (HAZ) until the weldment edge that is located away from the center point.

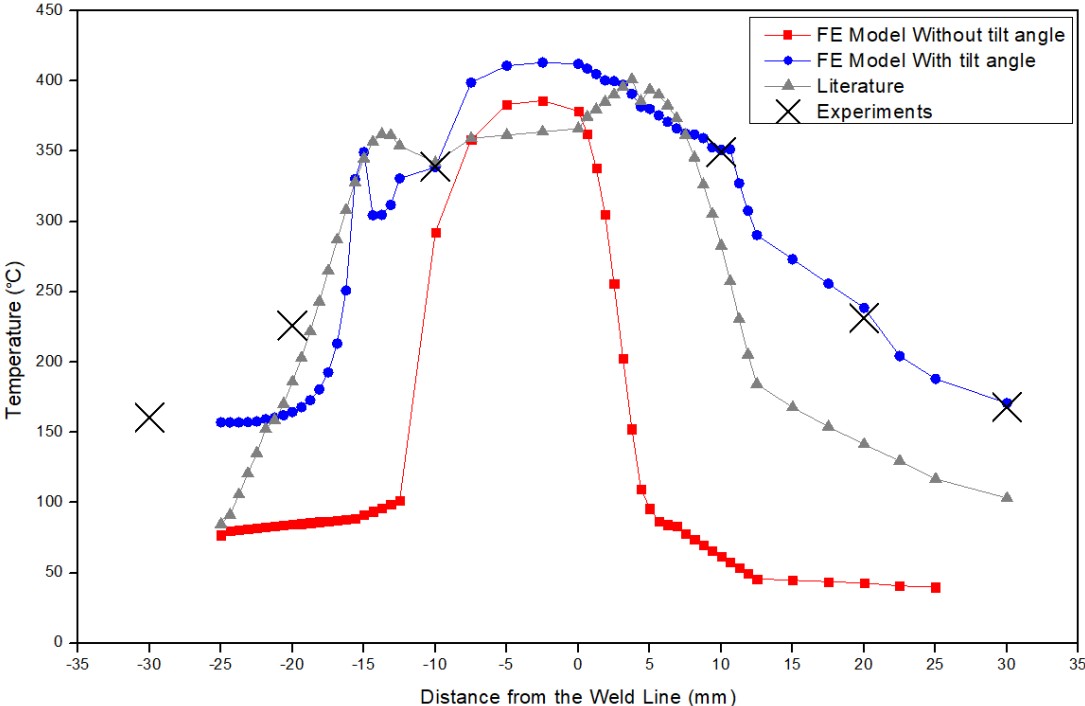

**Figure 20.** The cross-sectional view of the temperature with and without the tilt angle with comparison to the experiments and the literature [39].

The comparison between the models (with and without the tilt angle) and experiments [39] indicates that the maximum temperature increases with the increase of the tilt angle, because the peak temperature is faced by larger tangential movement at the tool rotating zone. To explain more, the maximum temperature inside the welding zone for the model with the tilt angle is around 413 °C at the welding shoulder area and it is around 300 °C at the pin root. In the model without the tilt angle, the peak temperature of 380 °C at the shoulder and the temperature of 250 °C at the pin root is obtained. The results illustrate that the peak temperature in the advancing side of the model with the tilt angle is in line with the experimental [39] observations and the literature [40].

Figure 20 also shows that the temperature has an "M"-shaped form, which is also labeled in the literature [41]. It should be noted that the literature [41] used the same process parameters. As can be seen in Figure 20 an upward trend for the temperature on both sides of the welding with a notable temperature variation on the advancing (around 350–400 °C) is seen in both numerical and experimental [39] observations. To clarify the point, due to the resistance of the transverse movement of the welding tool by the rotational movements of the shoulder and the pin, the increment in the temperature is more in the advancing side than the retreating side (temperature at the retreating side is around 300–350 °C) [42–46]. Thus, there is an asymmetrical pattern for the values of the temperature (higher temperature at the advancing side) with the maximum temperature of 413 °C. Another reason for this issue is non-uniformity of the flow of the material near the welding tool resulting in asymmetrical temperature distribution below the shoulder.

To sum up, the measured temperature values by the experiments [39] is observed to be lower than the calculated values by finite element model. As can be seen, the comparison between the presented model and the literature [6,34,38,40] shows a good agreement. Therefore, the proposed finite element model in this study is able to accurately predict the thermomechanical behavior of FSW process.

## 5. Conclusions

The influence of the tilted angle on thermomechanical behavior of similar friction stir welding (FSW) for aluminum 6061-T6 is studied in this paper. Previous studies assumed a thin symmetric shape for the tool shear layer, while after applying the tilt angle, the thickness and the shape of this layer will be significantly changed. Moreover, the tilt angle modification should be properly analyzed due to the reports on difficulties including complex thermomechanical behavior and mesh distortion. Thus, the Eulerian method is employed in this work because it can handle large changes in the material motion, and it is able to investigate the thermomechanical behavior during FSW.

- For the model without the tilt angle, the temperature at the tool is found to be around 873 °C; however, it gradually decreases in the direction of the welding transverse speed. Meanwhile, at the welding center point, the maximum temperature is around 389 °C, which is lower than the material melting temperature.
- It was reported that the nominal temperature of the area near the weldment specimen edge reaches 36 °C.
- Furthermore, the highest temperature was recorded at the upper surfaces of the intersection between the shoulder and the pin region, while the bottom surface of the workpiece recorded lower temperature.
- The cross section view also illustrates an almost non-symmetrical temperature distribution where the temperature at the advancing and retreating side is different. In the model with the tilt angle, the tool temperature was recoded to be around 1118 °C, while the temperature at the welding plate is around 413 °C, which is higher compared to the model without the tilt angle. It was shown that the maximum temperature is recorded at the welding zone, and it declines via the heat affected zone (HAZ) until the weldment edge moved away from the center.
- It was observed that the friction at the shoulder/sheets causes an asymmetrical distribution for the temperature in both X and Y direction. In the X direction, this issue is caused by the differences in the frictional force at the advancing and retreating sides while, in the Y direction, the issue is caused by the presence of the tilt angle. In the Y direction, the tilt angle also causes higher deformation area and higher mixing of the material.
- Moreover, the distributions of the strain rate have shown different behaviors because welding behavior is influenced by the impact of the tilt angle at different welding edges. This matter leads to the increase in the maximum strain rate in the joint section. Therefore, an improvement in the mixing of the material (at the back side of the tool) is seen. This improvement subsequently enhances the welding quality and decrease the likelihood for defects.

- Finally, after comparing the obtained results with the outcomes reported by the published studies, the validation and the verification of the results is proved.

**Author Contributions:** B.M. wrote the main manuscript and M.A. reviewed and revised it. All authors have read and agreed to the published version of the manuscript.

**Funding:** Institute of Transport Infrastructure at Universiti Teknologi PETRONAS with the cost center of 015NB0-001.

**Data Availability Statement:** Not Applicable.

**Acknowledgments:** The authors would like to acknowledge the Institute of Transport Infrastructure at Universiti Teknologi PETRONAS with the cost center of 015NB0-001. Moreover, the authors would like to thank Patricio Fernando Mendez, Amitava De, Wallace Kaufman and Reza Teimouri for their endless support.

**Conflicts of Interest:** All co-authors have seen and agree with the contents of the manuscript and there is no financial interest to report.

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
