# Peer review of "The Influence of the Tool Tilt Angle on the Heat Generation and the Material Behavior in Friction Stir Welding (FSW)"

_metals, doi:10.3390/met12111837_

Round 1

Reviewer 1 Report

The manuscript provided some excellent results of friction stir welding depending on the tool tilt angle. The manuscript is well-written and the reviewer was pleased with the smooth and continuous flow of the story. A wide range of literature including the state-of-the-art is reviewed to justify the need for this research. Aluminum 6061-T6 with a thickness of 6 mm was used as the base metal which is a very topical material because of its strength-to-weight properties. The process parameters were chosen as a rotational speed of 800 RPM, a transverse speed of 120 mm/min, and a plunging depth of 0.1 mm. However, the tilt angle used for the experiment is not clear. It was concluded that after considering the tilt angle, as the tool moves, a smooth and quick increase in the temperature at the tool's leading side achieves. It was also concluded that the model computational time is acceptable. The conclusion was drawn based on the experimental observation. The reviewer recommends accepting this article after some minor revisions.

1.  The abstract is too long with a lot of redundancies. Please reduce the abstract including the 3 P's (purpose, procedure, and principle findings) and 1C (putting the context). Lines 10-14 are redundant and remove those lines. Starts from In this study...

2. The following information should be included in the experimental section: "Finally, to investigate the in-113 fluence of the tilt angle on the intermediate variable of the FSW a comparison between the tilt angle of 0 ° and 1.5 ° is done."

3. It is also apparent that the experimental results were also obtained from the literature. In that case, figure 20 must be redrawn. Is the temperature profile obtained from the literature from numerical investigation or experimental investigation? What was the parameter used in the literature?  Are those the same? The result obtained from experiments is obtained from the reference [39]. Should not it be considered literature? Again the number of data points for this case is only 6. This is not convincing as the data points starts +- 10 mm which is very far from the weld center. The experimental data should include the weld zone and heat-affected zone. The reviewer believes that the inclusion of experimental data without the most crucial data points is not ideal. A few more data points must be included.

4. The reviewer cannot agree with the following statements "To sum up, the measured temperature values by the experiments [39] is a bit lower than the calculated values by finite element model (almost less than 5 percent difference),  while the difference between the presented models in the literature [6, 34, 38, 40] and the reality was around 20%" because of low data points. Another reason is those experiments were not conducted by the authors.

Author Response

Dear reviewer,

Thank you very much indeed for your many considerations and valuable remarks and comments.

Reviewer I

The manuscript provided some excellent results of friction stir welding depending on the tool tilt angle. The manuscript is well-written and the reviewer was pleased with the smooth and continuous flow of the story. A wide range of literature including the state-of-the-art is reviewed to justify the need for this research. Aluminum 6061-T6 with a thickness of 6 mm was used as the base metal which is a very topical material because of its strength-to-weight properties. The process parameters were chosen as a rotational speed of 800 RPM, a transverse speed of 120 mm/min, and a plunging depth of 0.1 mm. However, the tilt angle used for the experiment is not clear. It was concluded that after considering the tilt angle, as the tool moves, a smooth and quick increase in the temperature at the tool's leading side achieves. It was also concluded that the model computational time is acceptable. The conclusion was drawn based on the experimental observation. The reviewer recommends accepting this article after some minor revisions.

Thank you so much for your valuable remarks. More description about the tilt angle in experiment is added. Since, the experimental tests are only done to validate the simulated models, there is no separate part for the experiments in this paper. In the abstract, there was a mistake Leading side was a mistake and it was corrected and replaced by trailing side.

  1. The abstract is too long with a lot of redundancies. Please reduce the abstract including the 3 P's (purpose, procedure, and principle findings) and 1C (putting the context). Lines 10-14 are redundant and remove those lines. Starts from In this study...

Thank you for your valuable remark. The abstract is completely revised, Lines 10-14 were removed.

  1. The following information should be included in the experimental section: "Finally, to investigate the in-113 fluence of the tilt angle on the intermediate variable of the FSW a comparison between the tilt angle of 0 ° and 1.5 ° is done."

As mentioned in the first remark, since the experimental tests are only used to validate the simulation, there is no separated section for experiments in this study. However, the authors added an explanation regarding the experimental test in the paper body.

  1. It is also apparent that the experimental results were also obtained from the literature. In that case, figure 20 must be redrawn. Is the temperature profile obtained from the literature from numerical investigation or experimental investigation? What was the parameter used in the literature?  Are those the same? The result obtained from experiments is obtained from the reference [39]. Should not it be considered literature? Again the number of data points for this case is only 6. This is not convincing as the data points starts +- 10 mm which is very far from the weld center. The experimental data should include the weld zone and heat-affected zone. The reviewer believes that the inclusion of experimental data without the most crucial data points is not ideal. A few more data points must be included.

Both experimental and literature are used to verify and validate the results. At the center of the welding, we could not add some thermocouple and since our main concert was to improve the simulation and the contact area, the experimental study was limited.

  1. The reviewer cannot agree with the following statements "To sum up, the measured temperature values by the experiments [39] is a bit lower than the calculated values by finite element model (almost less than 5 percent difference),  while the difference between the presented models in the literature [6, 34, 38, 40] and the reality was around 20%" because of low data points. Another reason is those experiments were not conducted by the authors. have reviewed your paper titled: "The Influence of the Tool Tilt Angle on the Heat Generation and the Material Behavior in Friction Stir Welding (FSW) ".

The paragraph is completely revised and the values are corrected based on he reviewer valuable remark.

Reviewer 2 Report

Dear Authors,  

I have reviewed your paper titled: "The Influence of the Tool Tilt Angle on the Heat Generation and the Material Behavior in Friction Stir Welding (FSW) ".

The paper fulfils the aims and scope of Metals journal, and can be considered for potential publication. However, it needs some improvements. My overal merit about this work is high. I have some minor suggestions, which are listed below. 

General remarks:

- Please add the quantitative results into the abstract.

- You have presented 20 references. Only five have been published in last three years.  I suggest to support your work with newly published references more. It will increase the visibility of your work in scintific databases.

Introduction:

- This part is well described. Moreover, the novelty is clearly underlined.

- Lines 36-38 - I propose shortly describe, how these parameters influence on the mechanical properties of FSW joints.

Heat generation model descriptions:

- Please support presented equations sith relvant references.

Finite element model descriptions:

- Line 281 -  "The rotational velocity of 800 RPM and the transverse speed of 120 mm/min are selected for the welding." It is not clear, why these parameters were selected. Please describe in the paper.

- Please show mechanical properties of invetigated material. It will be easier to compare them to the results obtained in tests. Moroever, I propose to show chemical composition - readers will get full material description.

Results and Discussion:

- This part is very strong. Resuts have been analyzed and discussed very well.

- Fig. 11 - I propose o add values, it will be more clear.

- Fig. 19 - the same as above.

Conclusions:

- I propose to show the most important conclusions in points. It will be more readable for potential readers.

Author Response

Dear reviewers,

Thank you very much indeed for your many considerations and valuable remarks and comments.

Reviewer II

The paper fulfils the aims and scope of Metals journal, and can be considered for potential publication. However, it needs some improvements. My overall merit about this work is high. I have some minor suggestions, which are listed below. 

General remarks:

- Please add the quantitative results into the abstract.

Thank you for your comment. The abstract is completely revised and some values of the results are added to the abstract.

- You have presented 20 references. Only five have been published in last three years.  I suggest to support your work with newly published references more. It will increase the visibility of your work in scintific databases.

The references are revised and most recently published references are added.

Introduction:

- This part is well described. Moreover, the novelty is clearly underlined.

The novelty is highlighted.

- Lines 36-38 - I propose shortly describe, how these parameters influence on the mechanical properties of FSW joints.

More lines are added to explain more about the abovementioned parameters and the reason of selecting the parameters.

Heat generation model descriptions:

- Please support presented equations with relevant references.

As recommended by the reviewer, references for the equation are added.

Finite element model descriptions:

- Line 281 -  "The rotational velocity of 800 RPM and the transverse speed of 120 mm/min are selected for the welding." It is not clear, why these parameters were selected. Please describe in the paper.

The explanation are added in the paper to describe the reason of selecting the abovementioned parameters.

- Please show mechanical properties of investigated material. It will be easier to compare them to the results obtained in tests. Moreover, I propose to show chemical composition - readers will get full material description.

The mechanical properties of the material are highlighted, kindly see table 2 and table 3 as the material properties for aluminum 6061.

Results and Discussion:

- This part is very strong. Results have been analyzed and discussed very well.

Thank you very much indeed for your many considerations.

- Fig. 11 - I propose to add values, it will be more clear.

The values are added in the paper body, as recommended by reviewer, more values are added to describe the results more in detail.

- Fig. 19 - the same as above.

The values are added in the paper body, as recommended by reviewer, more values are added to describe the results more in detail.

Conclusions:

- I propose to show the most important conclusions in points. It will be more readable for potential readers.

As recommended by the reviewer, the conclusion is presented as bullet points.
